# Depression and Complicated Grief, and Associated Factors, of Bereaved Family Members of Patients Who Died of Malignant Pleural Mesothelioma in Japan

**DOI:** 10.3390/jcm11123380

**Published:** 2022-06-13

**Authors:** Yasuko Nagamatsu, Yumi Sakyo, Edward Barroga, Riwa Koni, Yuji Natori, Mitsunori Miyashita

**Affiliations:** 1Graduate School of Nursing Science, St. Luke’s International University, 10-1 Akashi-cho, Chuo-ku, Tokyo 104-0044, Japan; yumi-sakyo@slcn.ac.jp (Y.S.); edward-barroga@slcn.ac.jp (E.B.); 2St. Luke’s International Hospital, 9-1 Akashi-cho, Chuo-ku, Tokyo 104-8560, Japan; riwakoni@luke.ac.jp; 3Hirano Kameido Himawari Clinic, 7-10-1 Kameido, Koto-ku, Tokyo 136-0071, Japan; natori@himawari-clinic.jp; 4Department of Palliative Nursing, Health Sciences, Tohoku University Graduate School of Medicine, 2-1 Seiryomachi, Aoba-ku, Sendai 980-8575, Japan; miya@med.tohoku.ac.jp

**Keywords:** mesothelioma, grief, depression, complicated grief, asbestos, bereaved, family

## Abstract

Objectives: we investigated the prevalence and associated factors of depression and complicated grief (CG) among bereaved family members of malignant pleural mesothelioma (MPM) patients in Japan. Methods: Bereaved family members of MPM patients (*n* = 72) were surveyed. The Japanese version of the Patient Health Questionnaire-9 (PHQ-9) and the Japanese version of the Brief Grief Questionnaire (BGQ) were used to assess depression and complicated grief (CG), respectively. Socio-economic factors, anger toward asbestos, care satisfaction, achievement of good death, and quality of end-of-life care were assessed in relation to depression and CG. Results: In the family members of MPM patients, the frequencies of depression and CG were 19.4% and 15.3%, respectively. The bereaved family members who were not compensated by the asbestos-related health-damage relief system (*p* = 0.018) and who felt the financial impacts of the patient’s MPM on the family (*p* = 0.006) had a higher likelihood of depression. The bereaved family members who were not satisfied with the care given when the patient became critical (*p* = 0.034), who were not compensated by the asbestos-related health-damage relief system (*p* = 0.020), who felt the financial impact of the patient’s MPM on the family (*p* = 0.016), and whose deceased relative underwent surgery (*p* = 0.030) had a higher likelihood of CG. Conclusions: For bereaved family members of MPM patients, routine screening for depression and CG and the provision of grief care are suggested. In addition, for family members of MPM patients, financial support, including the promotion of the asbestos-related health-damage relief system, and improved care for patients who undergo surgery and when patients become critical, are recommended.

## 1. Introduction

Grief is a natural response to bereavement. The pain from grief usually eases gradually, and the bereaved eventually establish a new life without the deceased. However, some people experience ongoing poor psychological wellbeing, including depression and complicated grief (CG). CG is characterized by intense grief that lasts longer than usual and causes impairment in daily functioning [1]. It is important to be aware of the circumstances in which individuals may become more vulnerable to CG. One study in Japan found that CG occurred in 2.4% of the general population, and almost 25% when subclinical CG was included [2]. The prevalence of CG in bereaved family members of cancer patients was 14% [3]. The risk factors include place of death, inadequate social support, the family having difficulty accepting death, dissatisfaction with palliative care, perceived preparedness [4], and financial problems after death [1,5]. Additionally, a violent loss of life, such as suicidal death [6], death by terrorism [7], and homicide [8], is associated with a higher rate of CG. Other bereavement-related mental impairments, such as depression, may appear along with CG; however, they are considered independent, distinct entities [9].

Malignant pleural mesothelioma (MPM) is a rare, fatal malignancy caused by asbestos decades after the initial exposure [10]. Japan banned asbestos in 2006 and tightened regulations in 2012 [11]. People develop MPM not only by occupational exposure, but also by environmental exposure. An increased, scandalizing mortality ratio of mesothelioma in both sexes has been observed in Amagasaki city, which was the location of the major asbestos factories in Japan [12]. Occupational-oriented MPM is compensated by workmen’s accident compensation insurance, and environment-oriented MPM is compensated by the asbestos-related health-damage relief system [13]. The number of annual deaths caused by MPM in Japan is about 1600, and this number has been growing [14].

The survival period after the diagnosis of MPM is as short as 7–15 months [15,16,17,18]. MPM causes a series of debilitating symptoms [19,20], various emotional and psychological problems [21], and additional distress associated with legal procedures for compensation [22]. Furthermore, the family members of MPM patients are at risk of depression due to the impact of diagnosis [23] and may experience impaired emotional functioning [22] and caregiving burdens [24], which are risk factors for CG [1].

People with MPM reportedly receive little information about their disease, have a sense that their needs are ignored, and feel angry at their country and the employer responsible for their fatal disease [25], which impairs their quality of life [26]. This indicates that bereaved family members of MPM patients experience significant psychological distress. However, little is known about the psychological distress of the bereaved family members of MPM patients.

In this study, we aimed to investigate the prevalence and associated factors of depression and CG among the bereaved family members of MPM patients in Japan. The present study is part of a larger study on the quality of life of the bereaved family members of MPM patients.

## 2. Methods

### 2.1. Study Design, Participants, and Setting

A cross-sectional survey design was chosen to examine the prevalence and associated factors of depression and CG among the bereaved family members of MPM patients.

The inclusion criteria were people who (1) had lost a family member to MPM, (2) had a family member who had been diagnosed with MPM after 2008, when the first evidence-based chemotherapy succeeded in prolonging the survival of MPM patients, and (3) could answer a self-administered questionnaire written in Japanese. The exclusion criteria included bereaved family members who lost a family member within six months, as, according to a previous study, the diagnosis of CG should be made at least six months after the death of a family member [27]. This research is part of a larger study investigating the bereaved family members of MPM patients. The participants in this study were identical to the participants of a previously published study that investigated the achievement of a good death and quality of end-of-life care of MPM patients from the perspective of bereaved family members [28].

A request for cooperation was sent to the advocacy group of the Japan Association of Mesothelioma and Asbestos-Related Disease Victims and their Families. The association has 15 branches across Japan and works with approximately 700 victims of asbestos-related diseases and their families. The association generated the list of eligible bereaved family members according to the criteria and sent a set of the informed consent information and questionnaires to 109 eligible bereaved family members in November 2016. Those who agreed to participate returned the completed questionnaires via postal mail by March 2017.

### 2.2. Outcomes

The primary outcome was the prevalence of depression in bereaved family members of MPM patients. The secondary outcome was the prevalence of complicated grief in bereaved family members of MPM patients.

### 2.3. Instruments

#### 2.3.1. Information of the Patients and Bereaved Family Members

The following information was provided by the bereaved family members about the deceased patients: sex, age at diagnosis, survival, received treatments, and place of death. The receipts of two types of insurance compensation benefits were also obtained.

The information on the bereaved family members included the following: age, relationship to the patient, time of bereavement, experience of end-of-life discussion with the patient, timing of patient’s death, financial impact of patient’s MPM on family, and level of anger toward asbestos. The bereaved family members were also asked about their satisfaction with care upon diagnosis, when the patient became critical, and when the patient died.

#### 2.3.2. Depression

Depression was evaluated using the Japanese version of the Patient Health Questionnaire-9 (PHQ-9). The original PHQ-9 was developed to screen for depression, and its validity has been proven in several studies [29,30]. The PHQ-9 consists of nine items and is answered using a four-point Likert scale (0 = not at all, 1 = several days, 2 = more than half the days, 3 = nearly every day). PHQ-9 scores of 10 and over represented moderate to severe depression [31]. The meta-analysis by Manea et al. showed the sensitivity and specificity values of the PHQ-9 cutoff of ≥10 compared to semi-structured interviews are 0.88 and 0.86. The original PHQ-9 was translated into Japanese and validated with a Japanese population [32].

#### 2.3.3. Complicated Grief (CG)

CG was evaluated using the Japanese version of the Brief Grief Questionnaire (BGQ) [33], a validated Japanese version of the original BGQ developed by Shear [7] consisting of five items on CG to screen for CG. The items were answered using a three-point Likert scale (0 = not at all, 1 = somewhat, 2 = a lot), and the possible scores range from 0 to 10. A total score of 8 or higher on the BGQ indicates CG, between 5 and 7 implies probable CG, 5 or higher implies possible CG, and less than 5 denotes absence of CG [7]. In this study, bereaved family members who scored 9 or higher were considered to have CG.

#### 2.3.4. Achievement of Good Death (GDI)

The achievement of good death was assessed using the Good Death Inventory (GDI), which has been validated to evaluate the achievement of a good death from the perspective of bereaved family members [34]. The GDI consists of 18 items and is answered using a seven-point Likert scale (1 = absolutely disagree, 7 = absolutely agree). A high score suggests the achievement of good death.

#### 2.3.5. Quality of End-of-Life Care (CES)

The quality of end-of-life care was assessed by the short version of the Care Evaluation Scale (CES) [35]. The CES consists of 10 items. The bereaved family members answered using a six-point Likert scale (1 = highly disagree, 6 = highly agree). A higher score indicates better quality end-of-life care.

#### 2.3.6. Missing Data

Mean imputation was conducted for the missing data of the PHQ9, BGQ, GDI, and CES scores, according to the instructions for the tools.

### 2.4. Statistical Analysis

The scores of each scale were calculated under a scoring procedure. The scores of each item of the measurement scales (i.e., PHQ-9, BGQ, GDI, CES) were summed and used as the scale score.

First, we examined the presence of correlations between the total scores of the PHQ-9, BGQ, GDI, and CES. Then, the scores of the PHQ-9 and BGQ were examined with clinical social factors such as age and sex of patient and family member, survival, treatments received, place of death, approved compensations, experience of end-of-life discussion, satisfaction with care, financial impact of MPM on the family, timing of patient’s death, and level of anger towards asbestos (Appendix A).

Finally, we used the odds ratio and 95% confidence intervals (d) in binominal logistic regression analysis to assess the correlations between depression (PHQ-9 score was equal to or more than 10) and complicated grief (BGQ score was equal to or more than 8) and the clinical social factors. A *p*-value of <0.05 was considered statistically significant. Statistical analysis was performed using SPSS version 27.

### 2.5. Ethical Considerations

This study was approved by the Research Ethics Committee of St. Luke’s International University (16-A035). It was conducted based on the ethical principles of avoiding harm, voluntary participation, anonymity, and the protection of privacy and personal information.

## 3. Results

Of the 109 questionnaires distributed to the bereaved family members through the association, 74 (67.9%) were completed and returned. Two respondents who had experienced a loss within the past six months were excluded. Finally, a total of 72 questionnaires were subjected to analysis.

### 3.1. Characteristics of Malignant Pleural Mesothelioma Patients and Bereaved Family Members

As shown in Table 1, 81.9% of the deceased MPM patients were men, and their mean age at diagnosis was 66.9 years. The treatment modalities they received were chemotherapy (70.8%), palliative care (56.9%), and surgery (19.4%). A large minority (48.6%) died in the respiratory ward, followed by the PCU or hospice (33.3%). Only 13.9% died at home. The mean survival time was 14.5 months from the time of diagnosis. The majority of the bereaved family members (72.2%) were spouses of the MPM patients, and the mean bereavement time was 45.2 months.

### 3.2. Depression and Complicated Grief and among Bereaved Family Members

Of the 72 participants, 19.4% of the bereaved family members were screened as having moderate to severe depression. Based on the BGQ score, 15.3% suffered from CG and 56.9% exhibited probable CG. In total, 72.2% of the bereaved family members were categorized into possible CG (PCG). Two bereaved family members (2.8%) suffered from both depression and CG (Figure 1).

### 3.3. Correlation between the Total Scores of the PHQ-9, BGQ, GDI, and CES

The PHQ-9 score was significantly correlated with the BGQ score (r = 0.481, *p* = 0.000) but not with the GDI or CES scores. The BHQ score was significantly correlated with GDI (r = −0.403, *p* = 0.000), however, was not correlated with CES.

### 3.4. Factors Associated with Depression

The results of the binomial logistic regression analysis of depression are shown in Table 2. The bereaved family members who were not compensated by the asbestos-related health-damage relief system and who suffered a financial impact from the patient’s MPM had a higher risk of depression.

### 3.5. Factors Associated with BGQ Total Score

The results of the binominal logistic regression analysis for CG (BGQ score is equal to or more than 8) are shown in Table 3. The bereaved family members of deceased MPM patients who received surgery, whose households were financially impacted by MPM, who were not compensated by the asbestos-related health-damage relief system, and who were not satisfied with the care given when the patient became critical, were more likely to develop CG.

## 4. Discussion

This cross-sectional study demonstrated the prevalence of depression and CG among the bereaved family members of deceased MPM patients in Japan. The results showed: (1) the BGQ score and the PHQ-9 score were associated with GDI score; (2) depression and CG rarely occur at the same time in MPM; (3) financial impact and lack of compensation from the asbestos-related health-damage relief system are related to depression and CG; and (4) dissatisfaction with care when the patient became critical and received surgery are related to CG.

The rates of depression (19.4%) among family members of MPM patients were slightly higher, but almost at the same level, as reported for bereaved family members of other cancer patients, i.e., 15.5–17% [3,36]. Regarding CG, the rate of CG (BGQ ≥ 8) was 15.3%, which was higher than the 0.7–2.5% in the Japanese general population [2,37] and at the same level as the other cancer population (10.9–14%) [3,36] and cardio-vascular disease patients (14%) [38]. It was lower than the 61% for traffic accidents [39]. The possible CG (BGQ > 5) was 72.2%, which was higher than the Japanese general population at 2.5–22.7% [2,37] and the population of other cancers population at 55% [40]. The possible reasons for the high PCG in MPM are poor achievement of good death of the patient, unpreparedness and unacceptance of loss, and strenuous legal hurdles to claiming compensation for bereaved family members, who are often not compensated before the patient dies. A previous study showed some items of the GDI are related to CG [3]. In MPM, the GDI score was significantly poorer than in the wider cancer population [28]. Previous studies have also reported that advanced preparations for the loss [4] and acceptance of death [41] are associated with lower risks of bereavement-related complications. Unfortunately, MPM patients and their families generally have difficulty accepting the disease and facing death because MPM is caused by asbestos, and could have been avoided [25].

Another characteristic of grief in MPM is the low comorbidity of depression and CG. Only 2.8% of our sample had depression and CG at the same time. A systematic review by Komischke-Konnerg [42] estimated the co-occurrence of prolonged grief disorder and depression at 63%. The reason for the lack of co-morbid CG and depression in MPM is unclear, but the results of this study indicate that CG and depression are more distinguishable in MPM. A previous study reported that CG and depression can be considered as different forms of disorder, even though some of their symptoms overlap [43]. This may be related to the cause of distress. Ball et al. [44] reported that causes of psychological distress may differ in MPM and lung cancer because (1) MPM has a worse outlook than lung cancer, (2) there is additional stress due to legal and financial matters even after loss in MPM, and (3) MPM patients experience distress and blame a third party for the development of the disease.

The factors relating to depression and CG in MPM indicate that a lack of support impairs the quality of life of MPM patients, and, eventually, bereaved family members develop psychological distress; however, further research is necessary to prove this. Another important finding was that, in MPM, the financial impact on the household and the lack of compensation from the asbestos-related health-damage relief system related to both depression and CG. This finding supported previous studies reporting financial status as a factor related to depression [5] and CG [45] in the cancer population. Worker’s accident compensation insurance is more generous, but only available for occupational MPM. The current study showed that lack of compensation by the asbestos-related health-damage relief system that covers all MPM patients is associated with CG. However, financial impacts and lack of compensation from the asbestos-related health-damage relief system were independent related factors, meaning that even a recipient of compensation from the asbestos-related health-damage relief system may experience financial impacts. The results indicate that the compensation from the asbestos-related health-damage relief system may have a positive effect on bereaved family members, not only financially but also through easing the pain of victims. Further research is needed to clarify the effect of compensation on the bereaved family members of MPM patients, including whether compensation relieves the financial burden of affected families.

CG had additional related factors, such as patients undergoing surgery and dissatisfaction with care when the patient became critical. This finding suggests that the provision of quality care for MPM patients and their family before the patient’s death may be useful to prevent CG. The targeted points of care are when patients receive surgery and when the patient becomes critical. It is not clear how surgery is related to CG. The possible reasons may be complications [46], a reduction in lung volume after surgery [47], and reduced quality of life from pain [48]. As international guidelines recommend, surgery should be executed by skilled surgeons in high-volume centers, and should be considered only in a multimodality treatment plan for selected patients [49]. Other factors that have been reported to be associated with CG, such as the bereaved family member being female and the spouse of the deceased [50] and place of death [43], showed no significant association in the present study.

### 4.1. Implications of Care

Given the high prevalence of PCG in the current study, we recommend routine screening of depression and CG for bereaved family members of MPM patients. For those who have depression and CG, sufficient treatment must be provided by a specialist. Reportedly effective treatments should be considered, such as antidepressants for depression [51], and counseling [52] and cognitive behavioral therapy [53,54,55] for CG.

Care and social support obtained from a good support network were protective against depression and CG [42,56]. The recommended means highlighted in this study to support bereaved family members who suffer from depression and CG are financial support, including the promotion of the asbestos-related health-damage relief system; improvement in care for MPM patients, especially those who undergo surgery; and improvement in care when patients become critical.

### 4.2. Implications for Further Research

A future study to clarify the mechanisms of depression and CG among the bereaved family members of deceased MPM patients using multisite research across countries is recommended, as the number of family members of patients with MPM is limited in a single country. There is also a need to examine more psychosocial factors, such as posttraumatic stress disorder [57], pre-existing mental impairment [3], preparedness for death [58], and sense making [6]. Furthermore, the financial problems of MPM patients’ households and CG among bereaved family members of patients who undergo surgery need to be clarified to improve the quality of life of patients, and to prevent CG associated with MPM.

### 4.3. Representativeness of the General Population of Bereaved Family Members of MPM Patients

This study had a small convenience sample, as access to bereaved family members was limited because Japan has no registration system for people with MPM. Additionally, the bereaved family members assessed in this study were members of an advocacy group, so our results may not be representative of the general population of bereaved family members of deceased MPM patients. However, the characteristics of the patients of this study were similar to those in a previous study on MPM patients [26] and deceased MPM patients [16]. The majority were male [28] and over sixty years old. Around 20% underwent surgery [16], 70–80% received chemotherapy [28], around 20–30% received radiotherapy, and around 40% received palliative care. However, in this study, survival was 14.5 months, which is longer than average [16]. Furthermore, more patients in this study were compensated by the workmen’s accident compensation insurance (65%) and the asbestos-related health-damage relief system (78%) than previous studies (56% and 46%) [26].

### 4.4. Limitations

This study has some limitations. First, as we mentioned above, we had a small convenience sample. Second, the bereaved family members may have demonstrated recall bias because the mean duration of bereavement was 45 months. Finally, this study was a cross-sectional study. The results were based on self-report data, and no clinical interviews were conducted. We believe that loss of life caused by asbestos contributes greatly to the development of CG. To prove this hypothesis, more extensive studies with a larger number of participants are required. Specifically, a longitudinal study is warranted to develop an optimal support and care program.

## 5. Conclusions

The rates of depression and CG of bereaved family members of MPM patients were the same as for cancer and cardio-vascular disease and higher than in the general population but lower than it is for those affected by traffic accidents. PCG occurred more in MPM than in cancer. For bereaved family members, routine screening for depression and CG and the provision of grief care are recommended. In MPM, financial impacts and a lack of compensation from the asbestos-related health-damage relief system relates to both depression and CG, along with dissatisfaction with the care received when the patient becomes critical and undergoes surgery. These results suggest the importance of financial support for MPM patients and their family members, including the promotion of the asbestos-related health-damage relief system; improved care, especially for patients undergoing surgery; and improved care when patients become critical.

## Figures and Tables

**Figure 1 jcm-11-03380-f001:**
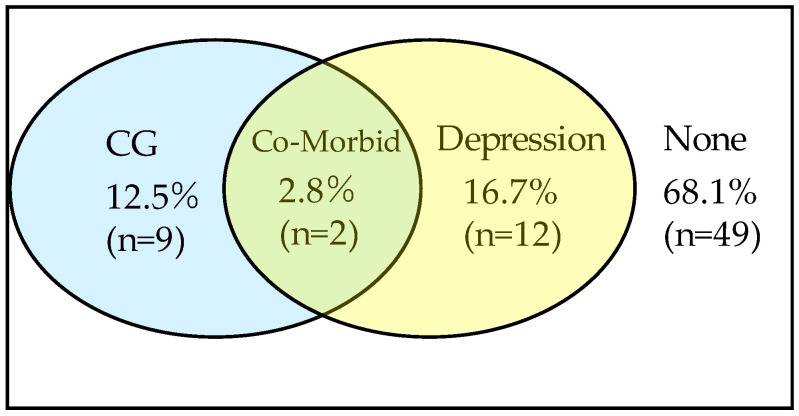
Percentage of complicated grief (CG) and depression in the bereaved family members.

**Table 1 jcm-11-03380-t001:** Characteristics of malignant pleural mesothelioma patients and their participating bereaved family members (*n* = 72).

Patients			*n*	%
Sex	Men		59	81.9
	Women		13	18.1
Source of asbestos exposure	Occupation		49	68.1
	Neighboring factory		17	23.6
	School		1	1.4
	Family		1	1.4
	Unknown		4	5.4
Treatment	Surgery		14	19.4
(includes multiple treatments)		Extrapleural pneumonectomy	12	16.7
		Pleurectomy decoration	2	2.8
	Chemotherapy		51	70.8
	Radiotherapy		15	20.8
	Palliative care		41	56.9
Compensation	Worker’s accident compensation insurance		47	65.3
(some had both types)	Asbestos-related health-damage relief system		56	77.8
Place of death	Respiratory ward		35	48.6
	Palliative care unit/hospice		24	33.3
	Home		10	13.9
	Other		3	4.2
Age at diagnosis (years)	Range:	36–92	Mean ± SD	66.9 ± 9.6
Survival (months)		0.5–69		14.5 ± 14.1
**Bereaved family members**			* **n** *	**%**
Sex	Men		15	20.8
	Women		57	79.2
Relationship with patient	Spouse		52	72.2
	Child		20	17.8
Experience of end-of-life discussion with patient	Yes		27	37.5
	No		44	61.1
Timing of patient’s death	Much sooner than expected		31	43.1
	Sooner than expected		25	34.7
	Moderate		9	12.5
	Later than expected		5	6.9
	Much later than expected		2	2.8
Satisfaction with care:	Satisfied		29	40.3
On diagnosis	Not satisfied		43	59.7
When patient became critical	Satisfied		31	38.9
	Not satisfied		41	61.1
When patient died	Satisfied		47	65.3
	Not satisfied		25	34.7
Financial impact of patient’s	Significant impact		12	16.7
MPM on family	Some impact		15	20.8
	Moderate impact		20	27.8
	Minor impact		15	20.8
	No impact		10	13.9
Level of anger toward asbestos	Very angry		56	77.8
	Angry		11	15.3
	Moderately angry		4	5.6
	Slightly angry		1	1.4
	Not angry at all		0	0
Age (in years)	Range:	32–82	Mean ± SD	62.5 ± 12.2
Time since bereavement (months)		9–110		45.2 ± 27.2

**Table 2 jcm-11-03380-t002:** Binominal logistic regression model predicting depression (*n* = 72).

Variable	Estimated Odds Ratio	95% CI	*p*-Value
Family financially impacted by patient’s MPM	2.569	1.316–5.015	0.006
Not compensated by the asbestos-related health-damage relief system	7.334	1.401–38.374	0.018

Model chi-square = 12.641, d = 1, *p* = 0.002, R^2^ = 0.263. Dependent variables: 1: PHQ-9 score is equal to or more than 10, 0: PHQ-9 score is less than 10.

**Table 3 jcm-11-03380-t003:** Binominal logistic regression model predicting CG (*n* = 72).

Variable	Estimated Odds Ratio	95% CI	*p*-Value
Family financially impacted by patient’s MPM	3.278	1.250–8.596	0.016
Not compensated by the asbestos-related health-damage relief system	19.210	1.609–229.392	0.020
Received surgery	11.301	1.256–101.649	0.030
Not satisfied with the care given when the patient became critical	13.626	1.213–153.009	0.034

Model chi-square = 22.206, d = 4, *p* = 0.001, R^2^ = 0.471. Dependent variables: 1: BGQ score is equal to or more than 8, 0: BGQ score is less than 8.

## Data Availability

The datasets generated and analyzed from this study are not publicly available to protect the anonymity of the participants, but are available from the corresponding author, Yasuko Nagamatsu, upon reasonable request.

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
