# Peer review of "Depression and Complicated Grief, and Associated Factors, of Bereaved Family Members of Patients Who Died of Malignant Pleural Mesothelioma in Japan"

_jcm, 2022, doi:10.3390/jcm11123380_

Round 1
Reviewer 1 Report
In this manuscript the authors describe a cross sectional survey study that examined the prevalence of depression (primary outcome) and complicated grief (secondary outcome) assessed with validated questionnaires, and explored 20 potentially associated factors (e.g., sex of patient, sex of respondent). Informed consent documents and surveys were sent to a convenience sample of family members of patients who had died of malignant pleural mesothelioma (MPM) more than 6 months previously and were members of an advocacy group for patients and family of MPM victims (n=109). Data from eligible respondents (n=72; 67.9% response rate) indicated that 19.4% of participants (n=14) had moderate to severe depression as assessed by their PHQ-9 score, while 15.3% (n=14) had complicated grief based on their BGQ score when assessed an average of 45.2 months after patient death. Associations were found (binomial logistic regression model) between the two outcomes. In addition, depression was associated with 1) family financially impacted by patient’s MPM; 2) Not compensated by the asbestos-related health damage relief system. In addition to associations with those two factors, complicated grief was also associated with: 3) received surgery; 4) not satisfied with care when the patient became critical.
Strengths of the study and manuscript include:
· The focus on the understudied areas of depression and complicated grief among bereaved family members of MPM patients
· A standardized procedure for identification, recruitment, and assessment of the bereaved family members asked to participate
· The use of validated questionnaires for collection of survey data on depression and complicated grief
· The use of binomial logistic regression to explore patient and family member factors associated with depression and complicated grief
· A thoughtful paragraph in the Discussion that highlights several major limitations of the study
The study and manuscript have several weaknesses that lessen enthusiasm for publication:
· Most critically, it is not clear that the sample included in the study is representative of the general population of bereaved family members of MPM patients in Japan. As the authors note in the Limitations paragraph (Section 4.3), the assessments were conducted with a relatively small sample of family members recruited through an advocacy group. While the authors note that recruiting a representative sample is made difficult by the absence of a registration system for people with MPM in Japan, more could be done to provide evidence that the sample included in the present study is representative. The authors’ own prior research (Nagamatsu et al, BMC Cancer, 2018) used a broader hospital-based approach along with an advocacy group approach to identify and recruit survivors of MPM to a survey study. Statistical comparison of demographic and clinical variables among the patients participating in that prior study to the same variables in the patients whose family members participated in the present study would be one way to support generalizability to the broader population of MPM patients.
· To better address the possibility of sampling bias even among those recruited through the advocacy group in the present study, the authors should indicate how the mailing list of bereaved family members contacted about the study (n=109 letters sent) was generated. Was only one family member per patient chosen to receive the invitation to participate? If so, how was that person selected? If more than one letter was sent per family, how was the final participant in the study decided upon? In addition, the authors should provide any data available that might allow comparison between the characteristics of those who agreed to be in the study and those who did not.
· Another substantial concern is the lack of attention paid to the distinction between clinical diagnoses and the experience of symptoms related to depression and complicated grief. The authors should indicate if the two questionnaires used in the study have been validated to provide cut points in the continuous scores generated by the questionnaires that would be consistent with clinical diagnosis. If so, then those cut points should be used for reporting prevalence and for the logistic regression analyses. If not, the authors should make it clear throughout the manuscript that severity of symptoms is being described.
· Given the cross-sectional, single sample, study design of the present study, the authors should avoid using any text that implies causal relationships and should clearly indicate when comparisons are being made (without statistical analysis) to previously published work by others (e.g., point #1 in the first paragraph of the Discussion).
· The authors should clearly indicate whether the participant sample in the present study is identical to, or substantially overlaps with, samples used in their previous published work (e.g., Nagamatsu et al, J Clin Med, 2022). If additional potentially relevant variables were assessed as part of the authors’ broader study, these should be included in the analyses for the present manuscript to explore relationships to depression and complicated grief.
Author Response
Point1. The authors’ own prior research (Nagamatsu et al, BMC Cancer, 2018) used a broader hospital-based approach along with an advocacy group approach to identify and recruit survivors of MPM to a survey study. Statistical comparison of demographic and clinical variables among the patients participating in that prior study to the same variables in the patients whose family members participated in the present study would be one way to support generalizability to the broader population of MPM patients.
Response 1:
Thank you for your valuable comments and suggestion. We have compared the characteristics of the MPM patients in the current study, our published study, and a study by Gemba. We added the explanation about representativeness of the general population in discussion section as follows;
“4.3. Representativeness of the general population of bereaved family members of MPM patients
This study had a small convenience sample, as access to bereaved family members was limited because Japan has no registration system for people with MPM. Additionally, the bereaved family members assessed in this study were members of an advocacy group, so our results may not be representative of the general population of bereaved family members of deceased MPM patients. However, the characteristics of the patients of this study were similar to those in a previous study on MPM patients [26] and deceased MPM patients [16]. The majority were male [28] and over sixty years old. Around 20% underwent surgery [16], 70–80% received chemotherapy [28], around 20–30% received radiotherapy, and around 40% received palliative care. However, in this study, survival was 13 months, which is longer than average [16]. Furthermore, more patients in this study were compensated by the workmen’s accident compensation insurance (65%) and the asbestos health damage relief system (78%) than previous studies (56% and 46%) [26]. “  P9
Point 2. To better address the possibility of sampling bias even among those recruited through the advocacy group in the present study, the authors should indicate how the mailing list of bereaved family members contacted about the study (n=109 letters sent) was generated. Was only one family member per patient chosen to receive the invitation to participate? If so, how was that person selected?
Response 2:
Thank you for your advice. According to the researchers’ request, the association generated the list of eligible bereaved family members and sent one questionnaire per one patient. We added the explanation as follows;
“A request for cooperation was sent to the advocacy group of the Japan Association of Mesothelioma and Asbestos-Related Disease Victims and their Families. The association has 15 branches across Japan and works with approximately 700 victims of asbestos-related diseases and their families. The association generated the list of eligible bereaved family members according to the criteria and sent a set of the informed consent information and questionnaires to 109 eligible bereaved family members in November 2016.” P3

Point 3: The authors should provide any data available that might allow comparison between the characteristics of those who agreed to be in the study and those who did not.
Response 3.
Thank you for your advice. Unfortunately, we could not the bereaved family members who did not answer us.
Point 4: Another substantial concern is the lack of attention paid to the distinction between clinical diagnoses and the experience of symptoms related to depression and complicated grief. The authors should indicate if the two questionnaires used in the study have been validated to provide cut points in the continuous scores generated by the questionnaires that would be consistent with clinical diagnosis. If so, then those cut points should be used for reporting prevalence and for the logistic regression analyses.
Response 4:Thank you very much an important advice.
1)In the instruments section (P3), we added the explanation about the scales, (1) developed for screening, (2) validity and cutoff score, and (3) usefulness in clinical setting as follows;
“2.3.2. Depression
Depression was evaluated using the Japanese version of the Patient Health Questionnaire-9 (PHQ-9). The original PHQ-9 was developed to screen for depression, and its validity has been proven in several studies [29][ 30]. The PHQ-9 consists of nine items and is answered using a four-point Likert scale (0 = not at all, 1 = several days, 2 = more than half the days, 3 = nearly every day). PHQ-9 scores of 10 and over represented moderate to severe depression [31]. The meta-analysis by Manea et al. showed the sensitivity and specificity values of the PHQ-9 cutoff of ≥10 compared to semi-structured interviews are 0.88 and 0.86. The original PHQ-9 was translated into Japanese and validated with a Japanese population [32].
2.3.3. Complicated Grief (CG)
CG was evaluated using the Japanese version of the Brief Grief Questionnaire (BGQ) [33], a validated Japanese version of the original BGQ developed by Shear [7] consisting of five items on CG to screen for CG. The items were answered using a three-point Likert scale (0 = not at all, 1 = somewhat, 2 = a lot), and the possible scores range from 0 to 10. A total score of 8 or higher on the BGQ indicates CG, between 5 and 7 implies probable CG, 5 or higher implies possible CG, and less than 5 denotes absence of CG [7]. In this study, bereaved family members who scored 9 or higher were considered to have CG.
2)We conducted the logistic regression analysis on two group according to the cutoff point.
Explanation is as follows ;
“Finally, we used the odds ratio and 95% confidence intervals (d) in binominal logistic regression analysis to assess the correlations between depression (PHQ-9 score was equal to or more than 10) and complicated grief (BGQ score was equal to or more than 8) and the clinical social factors. A p-value of <0.05 was considered statistically significant.” P4 Line23
3)We showed the prevalence of depression and CG according to the cut off score in the result section (P6) as follows;
“Of the 72 participants, 19.4% of the bereaved family members were screened as having moderate to severe depression. Based on the BGQ score, 15.3% suffered from CG and 56.9% exhibited probable CG.” P6 Line2
Point 5. Given the cross-sectional, single sample, study design of the present study, the authors should avoid using any text that implies causal relationships should clearly indicate when comparisons are being made (without statistical analysis) to previously published work by others (e.g., point #1 in the first paragraph of the Discussion)
Response 5:
Thank you so much for the important advice. We rephrased the text not to indicate the cause. 
1)We deleted the following sentence in the beginning of the discussion. P7 Line7
“(1) more PCG among bereaved family members of MPM patients than family members of other cancer patients,”
2)We rephrased the following sentences. P8 Line 9
The current study showed that an asbestos-related health damage relief system that covers all MPM patients helps reduce the risk of CG.  
⇒The current study showed that lack of compensation by the asbestos-related health damage relief system that covers all MPM patients is associated with CG.
3)We rephrased the following sentences. P8 Line 39
The recommended means highlighted in this study to mitigate the risk of depression and CG are financial support, including the promotion of the asbestos-related health damage relief system;
⇒The recommended means highlighted in this study to support bereaved family members who suffer from depression and CG are financial support, including the promotion of the asbestos-related health damage relief system;
Point 6. The authors should clearly indicate whether the participant sample in the present study is identical to, or substantially overlaps with, samples used in their previous published work (e.g., Nagamatsu et al, J Clin Med, 2022).
Response 6.
Thank you for your advice. We stated at the section(2.1 Study design, participants and setting;P2)that the participants are identical to our previous study as follows;
“The participants in this study were identical to the participants of a previously published study which investigated the achievement of a good death and quality of end-of-life care of MPM patients from the perspective of bereaved family members [28].” P2 Line50
Point 7. If additional potentially relevant variables were assessed as part of the authors’ broader study, these should be included in the analyses for the present manuscript to explore relationships to depression and complicated grief. 
Response 7.
Thank you for your advice. We analyzed the relationship between the scores of achievement of good death (Good Death Inventory :GDI ) and quality of end-of-life (the short version of the Care Evaluation Scale:(CES) that were reported in our previous study (Nagamatsu et al, J Clin Med, 2022) and PHQ-9 and BGQ scores. We added the GDI and CES in the instruments section (P3), statistical analysis (P4) and result(P6) and Discussion (P7) section as follows;
1)Instrument section
“2.3.4. Achievement of Good Death (GDI)
The achievement of good death was assessed using the Good Death Inventory (GDI), which has been validated to evaluate the achievement of a good death from the perspective of bereaved family members [34]. The GDI consists of 18 items and is answered using a seven-point Likert scale (1 = absolutely disagree, 7 = absolutely agree). A high score suggests the achievement of good death
2.3.5. Quality of end-of-life care (CES)
The quality of end-of-life care was assessed by the short version of the Care Evaluation Scale (CES) [35]. The CES consists of 10 items. The bereaved family members answered using a six-point Likert scale (1 = highly disagree, 6 = highly agree). A higher score indicates better quality end-of-life care.”
2) statistical analysis
“First, we examined the presence of correlations between the total scores of the PHQ-9, BGQ, GDI, and CES” P4 Line 16
3)Result
“3.3 Correlation between the total scores of the PHQ-9, BGQ, GDI, and CES.
The PHQ-9 score was significantly correlated with the BGQ score (r = 0.481, p = 0.000) but not with the GDI or CES scores. The BHQ score was significantly correlated with GDI (r=-0.403, p=0.000), however, was not correlated with CES.”
4)Discussion
“(1)the BGQ score and the PHQ-9 score were associated with GDI score;”
P7 Line 8
“The possible reasons for the high PCG in MPM are poor achievement of good death of the patient, unpreparedness and unacceptance of loss, and strenuous legal hurdles to claiming compensation for bereaved family members, who are often not compensated before the patient dies.” P7 Line 21
Reviewer 2 Report
Depression, anxiety, and complicated grief among family members with patients deceased by MPM in Japan is an interesting and original study. It would be best if the background on the general situation of grief and bereavement in Japan can be compared to this group with patients deceased by MPM would be necessary. The general situation of bereavement among Japanese family members should be described and how different are they different with this specific target group?
Readers will also need to know more about the occupational hazard prevention and compensation for workers in Japan. Without that, appreciation of policy recommendations can be serious being biased.
Author Response
Point .1 It would be best if the background on the general situation of grief and bereavement in Japan can be compared to this group with patients deceased by MPM. The general situation of bereavement among Japanese family members should be described and how different are they different with this specific target group?
Response 1:
Thank you for your valuable advice.
1)We stated the information about bereavement in the introduction as follows;
“One study in Japan found that CG occurred in 2.4% of the general population, and almost 25% when subclinical CG was included [2]. The prevalence of CG in bereaved family members of cancer patients was 14% [3].” P1 Line 44
2)In discussion, we made a comparison of grief between general population as well as other cause of death. P7 Line 13
“The rates of depression (19.4%) among family members of MPM patients were slightly higher, but almost at the same level, as reported for bereaved family members of other cancer patients, i.e., 15.5–17% [3] [36]. Regarding CG, the rate of CG (BGQ ≥8)was 15%, which was higher than 0.7% -2.5% in the Japanese general population [2][37] and at the same level as the other cancer population(10.9-14%) [3 ][36] and cardio vascular disease patients (14%) [38]. It was lower than 61% for traffic accidents [39]. The possible CG (BGQ > 5) was 72.2%, which was higher than the Japanese general population at 2.5%-22.7% [2][37][2] and the population of other cancer population at 55% [40]. “
Point 2. Readers will also need to know more about the occupational hazard prevention and compensation for workers in Japan. Without that, appreciation of policy recommendations can be serious being biased.
Response 2:
Thank you so much for the important advice.
1) In introduction, we added the explanation about asbestos control as well as the two types of compensation systems in Japan as follows (P2 Line11);
“Japan banned asbestos in 2006 and tightened regulations in 2012 [11]. People develop MPM not only by occupational exposure, but also by environmental exposure. An increased scandalized mortality ratio of mesothelioma in both sexes has been observed in Amagasaki city, which was the location of the major asbestos factories in Japan [12]. Occupational-oriented MPM is compensated by workmen’s accident compensation insurance, and environment-oriented MPM is compensated by the asbestos-related health damage relief system [13]. The number of annual deaths caused by MPM in Japan is about 1600, and this number has been growing [14].
2)In discussion we added the following sentences (P8 Line6).
“Worker’s accident compensation insurance is more generous, but only available for occupational MPM. The current study showed that lack of compensation by the asbestos-related health damage relief system that covers all MPM patients is associated with CG. However, financial impacts and lack of compensation from the asbestos-related health damage relief system were independent related factors, meaning that even a recipient of compensation from the asbestos-related health damage relief system may experience financial impacts. The results indicate that the compensation from the asbestos health damage relief system may have a positive effect on bereaved family members, not only financially but also through easing the pain of victims. Further research is needed to clarify the effect of compensation on the bereaved family members of MPM patients, including whether compensation relieves the financial burden of affected families.